# Depressive symptoms among adults is associated with decreased food security

**Shakila Meshkat[1], Hilary Pang[2], Vanessa K. Tassone [1,3], Reinhard Janssen-Aguilar[1], Michelle Wu[5], Hyejung Jung[5], Wendy Lou[5], Venkat Bhat [1,2,3,4] \***

1 Interventional Psychiatry Program, St. Michael's Hospital, Toronto, Ontario, Canada, 2 Department of Psychiatry, University of Toronto, Toronto, Ontario, Canada, 3 Institute of Medical Science, University of Toronto, Toronto, Ontario, Canada, 4 Mental Health and Addictions Services, St. Michael's Hospital, Toronto, Ontario, Canada, 5 Department of Biostatistics, Dalla Lana School of Public Health, University of Toronto, Toronto, Ontario, Canada

\* venkat.bhat@utoronto.ca

## Abstract

### Objective

We aim to evaluate the association of depressive symptoms, depressive symptoms severity and symptom cluster scores (i.e., cognitive-affective and somatic) with food security (FS). We will also evaluate the interaction effect of sex, income and ethnicity on these associations.

### Methods

Data from the 2005–2018 National Health and Nutrition Examination Survey cycles were used in this study. Participants included survey respondents 20+ years who had completed Depression and Food Security questionnaires. Multivariable logistic regression was used to estimate the associations between depressive symptoms and FS.

### Results

A total of 34,128 participants, including 3,021 (7.73%) with depressive symptoms, were included in this study. In both unadjusted and adjusted models, participants with depressive symptoms had lower odds of FS (aOR = 0.347, 95% CI: 0.307,0.391, p<0.001). Moreover, in both unadjusted and adjusted models, for each 1-point increase in cognitive-affective (aOR = 0.850, 95% CI = 0.836,0.864, p <0.001) and somatic symptoms (aOR = 0.847, 95% CI = 0.831,0.863, p <0.001), odds of high FS decreased correspondingly. Our study found no significant interaction effects of sex on depressive symptoms-FS association. Statistically significant interactions of ethnicity and poverty-to-income ratio on depressive symptoms-FS association were observed, revealing higher odds of FS among Non-Hispanic Black and Mexican American groups, and lower odds of FS in Non-Hispanic White and high-income subgroups.

**Data Availability Statement:** The data underlying the results presented in the study are available from https://wwwn.cdc.gov/nchs/nhanes/Default.

aspx. We used data from 2005-2006 through 2017-2018 survey cycle datasets.

**Funding:** The author(s) received no specific funding for this work.

**Competing interests:** I have read the journal's policy and the authors of this manuscript have the following competing interests: VB is supported by an Academic Scholar Award from the University of Toronto Department of Psychiatry and has received research support from the Canadian Institutes of Health Research, Brain & Behavior Foundation, Ontario Ministry of Health Innovation Funds, Royal College of Physicians and Surgeons of Canada, Department of National Defence (Government of Canada), New Frontiers in Research Fund, Associated Medical Services Inc. Healthcare, American Foundation for Suicide Prevention, Roche Canada, Novartis, and Eisai.

## Conclusion

Our study demonstrated an association between depressive symptoms and decreased FS. Further research is required to deepen our understanding of the underlying mechanisms and to develop focused interventions.

## 1. Introduction

Depression is a significant cause of disability on a global scale, impacting approximately 8% of adults in the United States (US) [1]. Individuals diagnosed with clinical depression often experience symptoms such as fluctuations in weight (either gain or loss), sleep disturbances (insomnia or hypersomnia), restlessness, persistent fatigue, a sense of worthlessness, diminished concentration, and thoughts of death or suicide [2]. Depression is not a singular condition but rather a combination of two distinct symptom clusters: somatic and non-somatic [1]. Research has demonstrated that analyzing these symptom clusters has advantages compared to solely examining overall depression [3]. Specifically, the cognitive-affective symptom cluster, involving thoughts and emotions, has been found to be linked to an individual's perception of unmet psychological care needs [4]. However, somatic symptoms do not share the same association with such needs [4].

Depression disproportionately affects individuals living in poverty, and women with household incomes below 100% of the federal poverty level (FPL) experience the highest rates of depression in the US [5]. Although depression and decreased FS are more prevalent among individuals living in poverty, there is evidence suggesting that decreased FS has an impact on mental health regardless of poverty level [6]. Studies have found that food insecure adults with household incomes at or below 130% of the FPL are more likely to experience depression compared to food secure adults within the same income range [7].

Food security (FS) refers to the state of consistently having access to an adequate amount of food to sustain a healthy and active lifestyle for all members of a household [8]. In 2018, a total of 14.3 million households in the US encountered instances of decreased FS [9]. Decreased FS is recognized as a significant health and nutrition concern in the US [9]. Adults experiencing decreased FS have higher rates of mental health issues and chronic diseases and they also tend to have lower scores on the Healthy Eating Index and consume more high-fat dairy products, sugary beverages, and salty snacks [10, 11]. Additionally, their intake of vegetables, fruits, dairy products, and essential vitamins like A, B-6, calcium, and magnesium is lower [12]. Several studies have examined the connection between decreased FS and adult mental health, yielding inconsistent findings [13, 14]. Moreover, the interaction effect of sex, ethnicity and income on association between decreased FS and depression is not fully known. Also, the sample sizes of previous studies were small [13, 14].

To fill these knowledge gaps, this study aims to further explore how depressive symptoms, depressive symptoms severity and symptom cluster scores are associated with FS. We will also evaluate the interaction effect of sex, ethnicity and income on association of depressive symptoms and with FS. We hypothesize that individuals with depressive symptoms will have higher odds of decreased FS.

## 2. Methods

### 2.1. Study Population

The study used data combined from six National Health and Nutrition Examination Survey (NHANES) cycles (2005–2018). NHANES has a cross-sectional design and aims to assess the

nutritional and health status of the general population residing in 50 states of the US by merging information from physical examinations, biomarkers, and questionnaires. Participants are selected using a complex, multistage probability sampling method to ensure representation of the civilian, noninstitutionalized US population. NHANES is conducted by the US National Center for Health Statistics (NCHS), Center for Disease Control and Prevention (CDC) and is regularly updated every two years. All participants provided informed consent, and the study received approval from the ethics review board of the NCHS. Additional details about methods and sampling are presented on the CDC website: **https://wwwn.cdc.gov/nchs/nhanes/**. We included males and females older than 20 years old who completed the Mental Health—Depression Screener (items DPQ010 to DPQ090), Food Security questionnaire (FSQ).

## 2.2. Exposure variable

Assessment of depressive symptoms was conducted using the Patient Health Questionnaire-9 (PHQ-9), a widely used self-report questionnaire designed to assess depressive symptoms based on established criteria as outlined in the Diagnostic and Statistical Manual of Mental Disorders, Fourth Edition for Major Depressive Disorder [15]. Participants responded to each question with a rating on a scale ranging from 0 to 3, indicating the frequency of their symptoms over the preceding two weeks, where 0 represented "not at all" and 3 represented "experienced nearly every day." Scores below 10 indicated an absence of depressive symptoms, while scores of 10 or higher suggested the presence of depressive symptoms. Several studies have indicated the reliability and validity of the PHQ-9 [15]. Symptom clusters were determined by summing scores to questions within each cluster (questions 1, 2, 6, 7, and 9 for cognitive-affective and questions 3, 4, 5, and 8 for somatic) and were kept as continuous values.

## 2.3. Outcome variable

FS was evaluated using the FSQ which is a well-designed questionnaire based on the US Food Security Survey Module designed by the US Department of Agriculture. The scale consists of a total of 18 items, with 10 items intended for adults in the household and 8 items for children. Since our sample only included adults, the 10 relevant items were utilized. Each item was recorded using a dichotomous response format (yes/no) and assessed various experiences over the past 12 months. These experiences include scarcity of food, worry about running out of food, affordability of nutritious meals, hunger, and reduced food intake. Participants read statements such as "I worried whether my food would run out before I got money to buy more," "The food that I bought didn't last, and I didn't have enough money to get more food," and "I couldn't afford to eat balanced meals." Affirmative responses to the questions included "often true" or "sometimes true," "yes," and "almost every month" or "some months but not every month." Based on the 10 items, participants were categorized into the following groups: high FS (0 items endorsed), marginal FS (1–2 items endorsed), low FS (3–5 items endorsed), and very low FS (6–10 items endorsed). For the purpose of our study, those with high or marginal FS were categorized as food secure, and those with low or very low FS were categorized as decreased FS.

## 2.4. Study covariates

Participants provided self-reported information regarding their sociodemographics, which was then used as covariates to account for factors that could potentially influence the relationship between depressive symptoms and FS. These characteristics included age (continuous in years), sex (female or male), income (assessed using the poverty-to-income ratio [PIR] and categorized as either low or high income), race/ethnicity (classified as non-Hispanic white, non-

Hispanic black, Hispanic, and other race), the highest level of education attained (less than high school, high school diploma or general educational development, some college, or college graduate and above), marital status (categorized as married/living with partner or not married/living with partner [ie. widowed, divorced, separated, never married]), and household size (continuous).

### 2.5. Statistical analyses

All statistical analyses were performed using R v 4.2.2 and the package "survey" to account for MEC survey weights. MEC survey weights were used to account for the clustered sample design, oversampling, survey non-response, and post-stratification adjustments so that the results could be generalized to the US population. Survey weights were divided by the number of cycles included to account for the merging of cycles, as recommended by NHANES. Categorical variables were described as raw frequency and weighted percent in the study population demographic characteristics table, while continuous variables were described as weighted mean and standard deviation (*SD*). A chi-square test of independence was used to check for statistically significant (*p*-value ≤0.05) differences in categorical demographic characteristics among the depressive symptoms groups, and a t-test was used to test for differences in continuous variables. Multivariable logistic regression was used to investigate the association between FS and presence of depressive symptoms (yes/no), total PHQ-9 score, and somatic/cognitive-affective scores. Lastly, exploratory multivariable logistic regressions with an interaction between depressive symptoms and 1) sex, 2) ethnicity, 3) PIR were run to test for any moderating effects on the relationship between depressive symptoms and FS. Subsequent subgroup analyses were conducted for any significant interactions and significance was set using Bonferonni correction. Listwise deletion was used to handle missing data.

## 3. Results

### 3.1. Study sample

Data from a total of 70,190 individuals were collected from the 2005–2018 NHANES cycles. The study population consisted of 34,128 individuals (Fig 1), of which 3,021 (7.73%) had depressive symptoms, and 6,328 (13.89%) reported food insecurity. The mean age of the study population was 47.46 (SD = 16.96), and 17,399 (51.32%) were females. Table 1 presents the demographic characteristics for the study population and any significant differences among covariates.

### 3.2. Depressive symptoms and FS

Unadjusted and adjusted models showed a statistically significant relationship between depressive symptoms and FS (Table 2). Those with depressive symptoms showed a 65.3% decreased odds of high FS (adjusted odds ratio [aOR] = 0.347, 95% CI = 0.307,0.391, *p*-value <0.001). For each one point increase in total PHQ-9 score, there was a 9.4% decrease in odds of high FS (aOR = 0.906, 95% CI = 0.897,0.915, *p*-value <0.001).

### 3.3. Cognitive-affective/somatic scores and FS

Unadjusted and adjusted models showed statistically significant relationships between cognitive-affective symptoms and FS, and between somatic symptoms and FS (Table 3). There was a 15.0% decrease in odds of high FS (aOR = 0.850, 95% CI = 0.836,0.864, *p*-value <0.001) for each one point increase in cognitive-affective scores, and a 15.3% decrease in odds for one point increased in somatic scores (aOR = 0.847, 95% CI = 0.831,0.863, *p*-value <0.001).

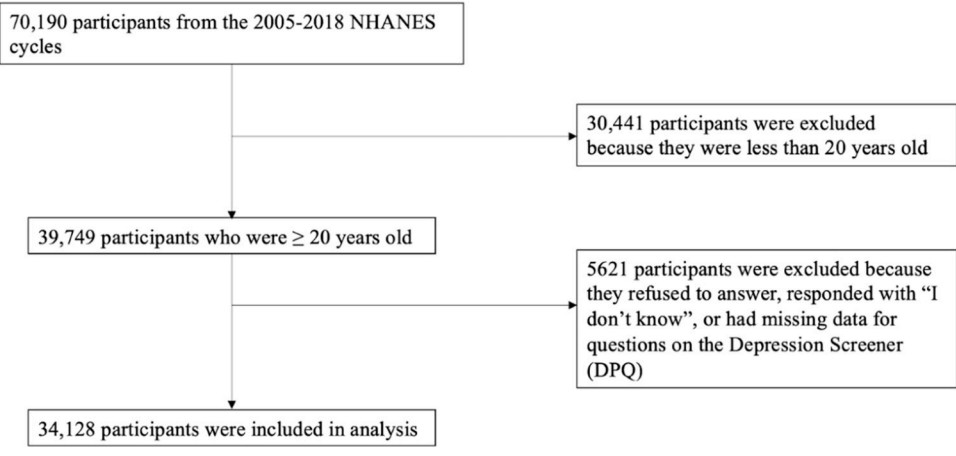

**Fig 1. Participant inclusion flowchart.**

### 3.4. Interaction effects of sex, ethnicity, PIR and depressive symptoms on FS

The interaction effect between sex and depressive symptoms was not significant on FS. Statistically significant results appeared in the interaction between some ethnicity groups and depressive symptoms, specifically in the Non-Hispanic Black*depressive symptoms (aOR = 1.61, 95% CI = 1.22, 2.13, $p$-value <0.001), and Mexican American*depressive symptoms (aOR = 1.55, 95% CI = 1.07, 2.24, $p$-value = 0.021), where Non-Hispanic White was the reference group. The interaction effect between PIR and depressive symptoms was also significant (aOR = 1.37, 95% CI = 1.09, 1.73, $p$-value = 0.009), where high income was the reference group.

**Table 1. Demographics of study sample.**

| Variable | Depressive symptoms (PHQ≥10) (n = 3021) | No depressive symptoms (PHQ<10) (n = 31,107) | p-value |
|---|---|---|---|
| Age (mean, SD) | 46.86 (15.83) | 47.52 (17.05) | 0.143 |
| Sex—Female (%) | 1919 (64.29) | 15480 (50.24) | <0.001 |
| PIR—High income (%) | 1294 (58.39) | 20209 (80.76) | <0.001 |
| Ethnicity (%) | | | <0.001 |
| Mexican American | 461 (8.15) | 4881 (8.40) | |
| Non-Hispanic Black | 674 (13.65) | 6662 (10.81) | |
| Non-Hispanic White | 1276 (63.44) | 13327 (68.25) | |
| Other—Hispanic | 387 (7.61) | 2854 (5.26) | |
| Other | 223 (7.14) | 3383 (7.29) | |
| Marital Status—Married/living with partner (%) | 1362 (47.79) | 19002 (64.93) | <0.001 |
| Education (%) | | | <0.001 |
| < High school | 1081 (25.56) | 7185 (14.69) | |
| High school | 730 (27.01) | 7157 (23.09) | |
| Some college | 902 (33.82) | 9234 (31.42) | |
| College+ | 305 (13.60) | 7508 (30.80) | |
| Household Size (mean, SD) | 2.98 (1.62) | 3.00 (1.52) | 0.417 |
| FS (%) | | | <0.001 |
| Secure | 1824 (64.59) | 25976 (87.92) | |
| Insecure | 1197 (35.41) | 5131 (12.08) | |

**Table 2. Unadjusted and adjusted ORs for associations between depressive symptoms and FS (depressive symptoms, PHQ-9 total score).**

| | OR (95% CI) | p-value | aOR* (95% CI) | p-value |
|---|---|---|---|---|
| Depressive symptoms (PHQ≥10) | 0.251 (0.223,0.282) | <**0.001** | 0.347 (0.307,0.391) | <**0.001** |
| Depressive symptoms (PHQ≥10)—Total Score | 0.887 (0.879, 0.895) | <**0.001** | 0.906 (0.897,0.915) | <**0.001** |

*aOR (adjusted odds ratios) control for age, sex, PIR, ethnicity, marital status, education, and household size

**Table 3. Unadjusted and adjusted ORs for associations between depressive symptoms and FS cognitive-affective/somatic scores.**

| | OR (95% CI) | p-value | aOR* (95% CI) | p-value |
|---|---|---|---|---|
| Cognitive-affective | 0.815 (0.802,0.827) | <**0.001** | 0.850 (0.836,0.864) | <**0.001** |
| Somatic | 0.821 (0.807, 0.835) | <**0.001** | 0.847 (0.831,0.863) | <**0.001** |

*aOR (adjusted odds ratios) control for age, sex, PIR, ethnicity, marital status, education, and household size

**Table 4. Unadjusted and adjusted ORs from logistic regression for the associations between depressive symptoms and FS, subgrouped by ethnicity and PIR.**

| | OR (95% CI) | p-value | aOR* (95% CI) | p-value |
|---|---|---|---|---|
| Depressive symptoms—Yes (Mexican Hispanic) | 0.43 (0.34,0.53) | <**0.001** | 0.46 (0.34,0.63) | <**0.001** |
| Depressive symptoms—Yes (Non-Hispanic Black) | 0.37 (0.32,0.44) | <**0.001** | 0.45 (0.37,0.54) | <**0.001** |
| Depressive symptoms—Yes (Non-Hispanic White) | 0.19 (0.16, 0.23) | <**0.001** | 0.31 (0.26,0.36) | <**0.001** |
| Depressive symptoms—Yes (Other) | 0.25 (0.16,0.37) | <**0.001** | 0.35 (0.22,0.55) | <**0.001** |
| Depressive symptoms—Yes (Other, Hispanic) | 0.38 (0.29,0.51) | <**0.001** | 0.44 (0.30, 0.63) | <**0.001** |
| Depressive symptoms—Yes (Low income) | 0.43 (0.37,0.49) | <**0.001** | 0.41 (0.36,0.47) | <**0.001** |
| Depressive symptoms—Yes (High income) | 0.25 (0.21, 0.30) | <**0.001** | 0.30 (0.25,0.36) | <**0.001** |

*aOR (adjusted odds ratios) control for age, sex, PIR, marital status, education, and household size, statistical significance p<0.0125

Subsequent ethnicity subgroup analysis showed that all subgroups with depressive symptoms had lower odds of FS, but those that were Non-Hisapnic White had the largest decrease in odds of FS (69% lower odds) compared to those with v who were Non-Hispanic Black (55% lower odds), or Hispanic Mexican (54% lower odds) (Table 4). The odds of being food secure were lower in those with depressive symptoms in both PIR subgroups, but those who were high-income showed a larger decrease in odds compared to those who were low-income (70% lower vs 59% lower) (Table 4).

## 4. Discussion

In this study, we used a nationally representative dataset to evaluate the associations of depressive symptoms, depressive symptoms severity and symptom cluster scores with FS. We also evaluated the interaction effect of sex, PIR and ethnicity in these associations. The results of our study indicated that participants with depressive symptoms have lower odds of being FS than those without depressive symptoms. A similar trend was seen for depressive symptoms severity, with those with more severe symptoms having lower odds of high FS. The analysis found no significant interaction effects between sex and depressive symptoms on FS. However, significant results were observed in the interaction between ethnicity and depressive symptoms, particularly in Non-Hispanic White and Non-Hispanic Black groups, where low FS risk was higher. Additionally, the interaction between PIR and depressive symptoms showed lower odds of FS, with high-income individuals showing a larger decrease in odds. Subgroup

analyses revealed that Non-Hispanic White individuals with depressive symptoms had the greatest decrease in odds of FS compared to other ethnicities, and high-income individuals with depressive symptoms showed a larger decrease in odds than those with low-income.

Our findings align with previous research that has shown a positive correlation between depression and decreased FS [16–18]. A meta-analysis conducted a comprehensive evaluation of 57 studies, involving a considerable participant sample of 169,433 individuals, offered compelling evidence supporting the association between decreased FS and depression [16]. Notably, the studies encompassed diverse populations, ranging from college freshmen, veterans, seniors, human immunodeficiency virus positive individuals, and low-income caretakers, to the general population, making it one of the largest and most inclusive investigations into mental health outcomes [16]. Our study, using NHANES dataset, provided a more nuanced exploration of the association between depression and decreased FS within this nationally representative sample. Despite a comprehensive meta-analysis demonstrating a positive correlation between decreased FS and depression across diverse populations [16], our study offers valuable insights that complement and extend this existing literature. Unlike the meta-analysis, which examined a broad range of populations, our study focused on the unique characteristics of individuals within the NHANES dataset. Furthermore, our study delves deeper into the modifying effects of sex, ethnicity, and income on the relationship between depression and FS. While the meta-analysis offered a broad overview, our study provides a more detailed analysis of these specific factors within this nationally representative sample. Of particular note is the discrepancy in findings related to the association between sex and the relationship between decreased FS and depression within the NHANES dataset. The meta-analysis suggested a lower association in women, whereas our study did not find significant differences between male and female participants. This contrasting result underscores the importance of exploring potential nuances within different population groups and highlights the need for further studies that consider these complexities. Reeder et al.'s study, using NHANES data, also highlighted the link between decreased FS and depressive symptoms [5]. However, their study did not delve into the severity of depressive symptoms or symptom clusters. Unlike their research, our study, with a larger sample size, examined these aspects and also explored the modifying effects of sex, ethnicity, and income [5]. As aforementioned, cognitive-affective and somatic symptoms of depression include negative mood and fatigue or loss of energy [4, 5]. Our findings indicated that both of these symptoms are associated with decreased FS, consistent with previous studies that reported positive association of fatigue and low mood with decreased FS [19, 20].

In our study, the odds of experiencing FS were lowest among non-Hispanic Whites, followed by non-Hispanic Blacks and Hispanic Mexican individuals with depressive symptoms. The complex interplay of factors contributing to this relationship involves a blend of socioeconomic, cultural, and systemic dynamics [21]. Historical privilege may have created expectations of stability within the non-Hispanic Whites population, making it harder for those struggling with mental health to seek assistance. This, coupled with the stigma around mental health issues, might deter individuals from seeking help, affecting their ability to maintain steady employment and subsequently access to food. In contrast, stronger community ties and mutual support within non-Hispanic Black communities might offer a safety net, mitigating the impact of depression on their FS. Our findings are inconsistent with previous research that indicated higher odds of decreased FS in Black Latino/Hispanic individuals [22, 23]. The link between ethnicity and FS is complex, as it is closely tied to other risk factors such as poverty and unemployment [22, 23]. Earlier research suggests that the heightened concentration of social and economic challenges among racial/ethnic minorities significantly contributes to their increased rates of decreased FS [23, 24]. The inconsistency in findings between our study

and most other studies could be attributed to the specific population being studied–individuals with depression. Depression itself can introduce a layer of complexity that may influence how decreased FS is experienced and reported [25]. Depression can affect appetite, motivation, and the ability to engage in activities of daily living, including accessing food [1, 2]. This could potentially lead to underreporting of decreased FS among those with depression, as they may not accurately recognize or communicate their struggles. Therefore, the unique psychological and physiological effects of depression could significantly alter the relationship between ethnicity and FS within this particular subset of the population, potentially explaining the differing results from broader studies. Several additional factors could explain this inconsistency. Firstly, regional variations and socioeconomic disparities may play a significant role. Different studies might focus on different geographic areas with varying levels of access to resources and support systems. Secondly, the specific criteria used to define decreased FS could differ across studies, leading to varying results. Additionally, the size and diversity of the samples in each study might influence the outcomes, as well as the time period during which the research was conducted. Overall, the complex interplay of these factors underscores the need for nuanced analysis when interpreting disparities in decreased FS among different ethnic groups, especially within specific population subsets like individuals with depression.

Our findings demonstrated higher odds of FS among adults with depressive symptoms and high income compared to those with low income. The association between high income and FS is well-established [26, 27]. Generally, individuals with higher incomes tend to have better access to a diverse range of nutritious foods, as they have the financial means to purchase them. This often results in improved dietary quality and reduced risk of decreased FS [26]. Higher income households are more equipped to handle unexpected expenses or fluctuations in food prices, ensuring a consistent food supply [26, 27]. However, it's important to note that while high income can contribute to FS, other factors like employment stability, access to education, and health and social policy also play significant roles in ensuring a population's overall FS. In addition, the nutritional content of the food supply is important as it has been previously shown that greater proximate access to vegetables and fruits moderated food insecurity's association with poor mental health, though its association with depression specifically has not yet been studied [28]. Interestingly, political decision-making and expansion of corporate food retailers can play an important role in FS. FS is impacted by food accessibility, the geographical distance to a food retailer, such as traditional supermarkets and more recently, dollar stores. Although it has been suggested that the presence and entry of dollar stores are not more likely to be associated with areas categorized as "low-income and low-access", once a dollar store enters a food desert, that area is more likely to remain without access to a supermarket [29]. Dollar stores have been suggested to be filling a gap in food access in places where no other food retailer will enter and are more likely to expand into areas with severe food access limitations compared to traditional supermarkets [29]. Future studies can also examine the association between the macroeconomic conditions as well as rural vs. urban setting on FS and depression.

Conducting a thorough assessment of the link between depression and FS is essential. Regardless of the order of onset, the concurrent presence of these issues inevitably exacerbates individuals' health and adversely affects their households' circumstances. As a result, addressing this intersection becomes paramount for effective treatment strategies. Among individuals with both mental disorders and decreased FS, there is a notable increase in hospitalization and utilization of mental health services, indicating a significant burden of mental illness within this group affected by decreased FS [30]. The connection between decreased FS and mental health service utilization shows that greater levels of decreased FS are linked to increased use of mental health services [28]. This situation raises concern due to the substantial burdens

posed by each condition independently. Individuals facing both decreased FS and depression form a particularly vulnerable group that requires focused attention from policy and health service interventions. Physicians who are responsible for diverse or marginalized patient populations should consider conducting additional screening for depression for individuals who test positive for decreased FS. Similarly, for patients showing signs of depression, it might be relevant to assess the presence of decreased FS, particularly due to the known connection between nutrition and emotional well-being. This approach aims to ensure comprehensive care and address potential interrelated health issues in these vulnerable groups.

Several pathways can explain the association between depressive symptoms and decreased FS. Decreased FS leads to inadequate nutrition, which often prompts individuals to opt for cheaper, less nutritious foods high in fat and carbohydrates but low in essential vitamins and micronutrients [10, 11]. This nutritional deficiency is linked to a higher risk of depression [12]. Prolonged malnutrition can negatively impact mental health, particularly since antioxidants such as vitamin C and vitamin E, along with other carotenoids, have protective effects against depression, and their deficiency could contribute to its development [31]. Furthermore, decreased FS can increase stress levels, particularly stress related to providing food for oneself or one's family, which is associated with increased risk of depression [32]. Notably, individuals with low socioeconomic status or living in poverty often exhibit higher levels of stress hormones such as cortisol and epinephrine [33]. However, this association between decreased FS and depression appears to persist independent of income levels [34]. Besides, decreased FS likely amplifies feelings of shame, which itself has been connected to a heightened risk of depression [35]. Lastly, depression can hinder an individual's ability to secure and maintain employment, especially among those with lower education levels, potentially exacerbating their risk of decreased FS [36].

Our study has both strengths and limitations. One of its strengths lies in the extensive and nationally representative nature of the NHANES dataset, allowing for a comprehensive assessment of the topic. Additionally, NHANES employs rigorous sampling and data collection processes carried out by trained interviewers, enhancing the credibility of the findings. However, this study also faces certain limitations. It is essential to acknowledge that the FS data in the NHANES dataset are self-reported, which introduces potential recall bias and desirability bias. Additionally, the FSQ questionnaire focuses on meals such as breakfast, lunch, and dinner, excluding snacking occasions. Moreover, the study's cross-sectional design may hinder the establishment of causal relationships between depressive symptoms and FS, necessitating caution in interpreting the results. Despite these limitations, the study contributes to the understanding of the complex interplay between food-related behaviors and depressive symptoms.

In conclusion, this study utilizing NHANES dataset has shed light on a significant association between depressive symptoms and decreased FS. The findings underscore the pressing need to address both mental health and nutritional concerns as interconnected public health challenges. Future studies with large sample sizes are required to further understand the mechanism underlying this phenomenon.

## Author Contributions

**Conceptualization:** Shakila Meshkat, Venkat Bhat.

**Data curation:** Michelle Wu.

**Formal analysis:** Michelle Wu, Hyejung Jung.

**Investigation:** Shakila Meshkat, Hilary Pang, Michelle Wu, Venkat Bhat.

**Methodology:** Shakila Meshkat, Hilary Pang, Michelle Wu, Venkat Bhat.

**Project administration:** Shakila Meshkat, Venkat Bhat.

**Supervision:** Wendy Lou.

**Validation:** Shakila Meshkat, Venkat Bhat.

**Visualization:** Shakila Meshkat, Venkat Bhat.

**Writing – original draft:** Shakila Meshkat, Hilary Pang, Vanessa K. Tassone, Venkat Bhat.

**Writing – review & editing:** Shakila Meshkat, Hilary Pang, Vanessa K. Tassone, Reinhard Janssen-Aguilar, Michelle Wu, Hyejung Jung, Wendy Lou, Venkat Bhat.

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
