## [Decision Letter · Decision Letter 0]

26 Feb 2024

PONE-D-24-00420Depressive Symptoms Among Adults is Associated with Decreased Food SecurityPLOS ONE

Dear Dr. Bhat,

Thank you for submitting your manuscript to PLOS ONE. After careful consideration, we feel that it has merit but does not fully meet PLOS ONE’s publication criteria as it currently stands. Therefore, we invite you to submit a revised version of the manuscript that addresses the points raised during the review process.

1)  Article from Reeder et al looked at same NHANES database from 2005-2016 and reported that FS and depression are associated. Authors should mention this in the discussion section? Maybe noting that this study is similar but much larger?

Authors cite this article (reference number 5), talking about cognitive symptoms of depression and while talking about relationship between income and depression. But authors do not mention that reference number 5 looked at the same database as this study, similar time periord wihle looking at FS and depression 

2) In the discussion section, authors mention a much bigger study (170,000 patients) showing similar results. Maybe authors can briefly report how their study is important despite the earlier larger study showing similar results and how it adds to the literature Also explain how current study is different from the much bigger metanalysis showing similar results?

3) In the discussion section, Authors should talk about a possible explanation (even if it's obvious) why FS and depression are associated with each other. Currently in the discussion section, authors just reported that "People with depression have higher odds of lower FS" without talking about the reason why?. Especially as latest research notes that FS is related to depression irrespective of poverty level.

We look forward to receiving your revised manuscript.

Kind regards,

Vikramaditya Samala Venkata

Academic Editor

PLOS ONE

Reviewers' comments:

Reviewer's Responses to Questions

**Comments to the Author**

1. Is the manuscript technically sound, and do the data support the conclusions?

Reviewer #1: Yes

Reviewer #2: Yes

2. Has the statistical analysis been performed appropriately and rigorously? 

Reviewer #1: Yes

Reviewer #2: Yes

3. Have the authors made all data underlying the findings in their manuscript fully available?

Reviewer #1: Yes

Reviewer #2: Yes

4. Is the manuscript presented in an intelligible fashion and written in standard English?

Reviewer #1: Yes

Reviewer #2: Yes

5. Review Comments to the Author

Reviewer #1: The large sample size and robust methodology strengthen the credibility of the findings. Acceptance of this abstract is recommended as it significantly contributes to the understanding of the complex interplay between mental health and food security, calling for further exploration and targeted interventions.

Reviewer #2: The authors did great job conducting this study. This study brings great insights about the association between food security and depression, which is consistent with literature. It highlights the importance of food security, which can be topic of discussion for clinicians while treating patients with depression. The authors used good research tools. The statistical analysis looks good. The conclusion could be stronger with heading.

6. PLOS authors have the option to publish the peer review history of their article (what does this mean?). If published, this will include your full peer review and any attached files.

Reviewer #1: **Yes: **Ripudaman Munjal

Reviewer #2: **Yes: **Ram Kishun Verma

---

## [Author Response · Author response to Decision Letter 0]

3 Apr 2024

Response to reviewers 

1) Article from Reeder et al looked at same NHANES database from 2005-2016 and reported that FS and depression are associated. Authors should mention this in the discussion section? Maybe noting that this study is similar but much larger?

Authors cite this article (reference number 5), talking about cognitive symptoms of depression and while talking about relationship between income and depression. But authors do not mention that reference number 5 looked at the same database as this study, similar time periord wihle looking at FS and depression 

Response: Thank you for your comment. We added ‘’Reeder et al.'s study, using NHANES data, also highlighted the link between decreased FS and depressive symptoms. However, their study did not delve into the severity of depressive symptoms or symptom clusters. Unlike their research, our study, with a larger sample size, examined these aspects and also explored the modifying effects of sex, ethnicity, and income.’’ to the discussion section of the revised manuscript.

2) In the discussion section, authors mention a much bigger study (170,000 patients) showing similar results. Maybe authors can briefly report how their study is important despite the earlier larger study showing similar results and how it adds to the literature Also explain how current study is different from the much bigger metanalysis showing similar results?

Response: Thank you for your comment. We revised discussion section of the manuscript: ‘’Our study, using NHANES dataset, provided a more nuanced exploration of the association between depression and decreased FS within this nationally representative sample. Despite a comprehensive meta-analysis demonstrating a positive correlation between decreased FS and depression across diverse populations, our study offers valuable insights that complement and extend this existing literature. Unlike the meta-analysis, which examined a broad range of populations, our study focused on the unique characteristics of individuals within the NHANES dataset. Furthermore, our study delves deeper into the modifying effects of sex, ethnicity, and income on the relationship between depression and FS. While the meta-analysis offered a broad overview, our study provides a more detailed analysis of these specific factors within this nationally representative sample. Of particular note is the discrepancy in findings related to the association between sex and the relationship between decreased food security and depression within the NHANES dataset. The meta-analysis suggested a lower association in women, whereas our study did not find significant differences between male and female participants. This contrasting result underscores the importance of exploring potential nuances within different population groups and highlights the need for further studies that consider these complexities.’’

3) In the discussion section, Authors should talk about a possible explanation (even if it's obvious) why FS and depression are associated with each other. Currently in the discussion section, authors just reported that "People with depression have higher odds of lower FS" without talking about the reason why?. Especially as latest research notes that FS is related to depression irrespective of poverty level.

Response: Thank you for your response. We added the following paragraph to the discussion section regarding the mechanism underlying FS and depression.

‘’Several pathways can explain the association between depressive symptoms and decreased FS. Decreased FS leads to inadequate nutrition, which often prompts individuals to opt for cheaper, less nutritious foods high in fat and carbohydrates but low in essential vitamins and micronutrients. This nutritional deficiency is linked to a higher risk of depression. Prolonged malnutrition can negatively impact mental health, particularly since antioxidants such as vitamin C and vitamin E, along with other carotenoids, have protective effects against depression, and their deficiency could contribute to its development. Furthermore, decreased FS can increase stress levels, particularly stress related to providing food for oneself or one's family, which is associated with increased risk of depression. Notably, individuals with low socioeconomic status or living in poverty often exhibit higher levels of stress hormones such as cortisol and epinephrine. However, this association between decreased FS and depression appears to persist independent of income levels. Besides, decreased FS likely amplifies feelings of shame, which itself has been connected to a heightened risk of depression. Lastly, depression can hinder an individual's ability to secure and maintain employment, especially among those with lower education levels, potentially exacerbating their risk of decreased FS.’’

---

## [Editor Report · Decision Letter 1]

24 Apr 2024

Depressive Symptoms Among Adults is Associated with Decreased Food Security

PONE-D-24-00420R1

Dear Dr. Bhat,

We’re pleased to inform you that your manuscript has been judged scientifically suitable for publication and will be formally accepted for publication once it meets all outstanding technical requirements.

Kind regards,

Vikramaditya Samala Venkata

Academic Editor

PLOS ONE
---

## [Editor Report · Acceptance letter]

9 May 2024

PONE-D-24-00420R1 

PLOS ONE

Dear Dr. Bhat, 

I'm pleased to inform you that your manuscript has been deemed suitable for publication in PLOS ONE. Congratulations! Your manuscript is now being handed over to our production team.

Kind regards, 

on behalf of

Dr. Vikramaditya Samala Venkata 

Academic Editor

PLOS ONE